# Disgust in anorexia nervosa: Testing a theoretical model connecting negative body image to disgust propensity, disgust sensitivity, and self-disgust

Fleur Boonstra[1]*, Peter J. de Jong[1], Rebecca Schulz[1], Klaske A. Glashouwer[1,2]

**1** Department of Clinical Psychology and Experimental Psychopathology, University of Groningen, Groningen, The Netherlands, **2** Department of Eating Disorders, Accare Child and Adolescent Psychiatry, Groningen, The Netherlands

* f.boonstra@rug.nl

## Abstract

Consistent with the view that disgust is involved in the persistence of eating disorder symptomatology, it has been found that disgust propensity is related to a negative body image. Importantly, earlier research in non-clinical samples provided preliminary evidence that this relationship could be statistically accounted for by self-disgust. The current study tested the robustness of this finding and examined if this pattern would also be evident when including individuals with and without clinically diagnosed anorexia nervosa (AN). In addition, we tested whether the relationship between self-disgust and negative body image would be especially pronounced in individuals with high disgust sensitivity. Finally, we explored whether body checking and body avoidance could statistically account for the relationship between self-disgust and negative body image. To test these hypotheses, female adolescents with ($n = 64$) and without ($n = 62$) AN diagnosis completed questionnaires administered online. Results showed that (1) the relationship between disgust propensity and negative body image could be statistically accounted for by self-disgust; (2) disgust sensitivity did not moderate the relationship between self-disgust and negative body image, and; (3) the relationship between self-disgust and negative body image could be statistically accounted for by body checking, but not by body avoidance. Together, these findings are consistent with the view that self-disgust may be an important factor in the persistence of a negative body image in anorexia nervosa.

## Introduction

Anorexia nervosa (AN) is an eating disorder characterized by restrictive food intake, an intense fear of gaining weight, and a disturbance in the experience of one's body weight or shape [1]. Around 1–4% of women and 0.2–0.7% of men are affected by AN during their lifetime [2,3] and the disorder involves a high psychological, social,

**Data availability statement:** Yes - all data are fully available without restriction; All data underlying the findings reported in this manuscript are publicly available from Dataverse at: https://dataverse.nl/dataset.xhtml?persistentId=doi:10.34894/AULILF.

**Funding:** The author(s) received no specific funding for this work.

**Competing interests:** The authors have declared that no competing interests exist.

and economic impact on patients, their family, as well as society in general [4]. Almost half of individuals with AN do not improve after treatment, and even after initially successful treatment, the relapse rate is high [5–8]. This points to the need for a better understanding of the mechanisms underlying the core features of AN, thereupon more effective treatments can be developed that result in sustained recovery [9,10].

During the last two decades, several researchers have proposed that disgust and disgust-related mechanisms might be involved in AN [11–18]. Disgust is characterized by intense aversive feelings of revulsion, a distinctive facial expression, and a strong urge to distance oneself from the disgust-evoking stimulus [19]. Disgust is assumed to serve as a disease-avoidance mechanism by preventing infection from pathogens that are omnipresent but invisible to the naked eye [20,21]. Individuals differ in their habitual tendency to respond with disgust to stimuli or situations (i.e., *disgust propensity*; [22]). High disgust propensity has been linked to various psychopathologies [23,24]. Previous research emphasized the importance of distinguishing disgust propensity from *disgust sensitivity*, which is the extent to which someone negatively evaluates the emotion of disgust or is emotionally affected by feeling disgusted [22]. Increased levels of disgust sensitivity have been observed in individuals with AN compared to non-clinical groups [11,25–28].

Not only can disgust be elicited by external objects, but also by the self [29,30]. *Self-disgust* refers to the appraisal of physical or characterological/behavioral parts of the self as revolting. When individuals experience their own shape and weight as too fat, the confrontation with the own body can elicit intense disgust [31]. In the present study, when mentioning self-disgust, we are specifically referring to such physical/body-related self-disgust. Individuals with high levels of self-disgust, also show higher levels of both disgust propensity and disgust sensitivity on average [17,32,33]. In addition, individuals with AN reported experiencing disgust towards the own body and linked their experience to avoidance behaviors [34]. Moreover, individuals with AN reported higher self-disgust on questionnaires than individuals without an eating disorder [25,27,28].

On the basis of the available evidence, we previously proposed a theoretical model (Fig 1) to help explain how different forms of disgust may contribute to the development and persistence of a negative body image. Negative body image is a transdiagnostic feature of many eating disorders and has been associated with the development, maintenance and relapse in AN [36–38]. Body image is defined as a complex construct encompassing thoughts, behaviors, feelings and evaluations related to the own body [39]. A negative body image may manifest itself by a strong importance of, as well as preoccupation and dissatisfaction with the own shape and weight. Following our theoretical model, in response to social pressures and/or aversive experiences, individuals with a strong tendency to experience disgust (*disgust propensity*) may also be more likely to experience disgust towards their own body (i.e., *self-disgust*). Thus, high disgust propensity may render people susceptible to develop a stable appraisal of the own body as disgusting, which, in turn, might contribute to the development and reinforcement of a negative body image.

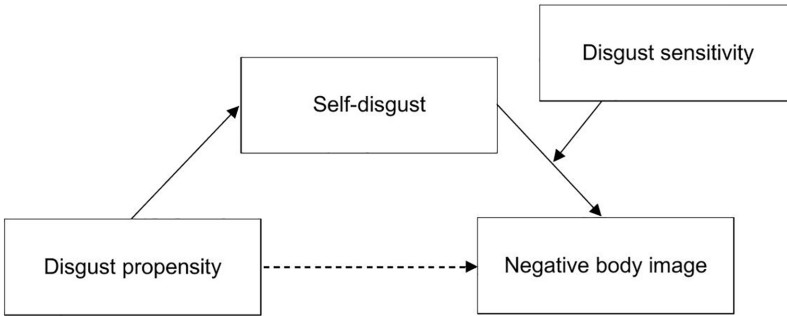

**Fig 1. Theoretical model adapted from von Spreckelsen et al. (2018) [35].**

In line with the latter idea, we found that self-disgust and negative body image indeed were positively related [35,40,41]. Furthermore, we propose that the relationship between self-disgust and negative body image might be moderated by *disgust sensitivity*. More specifically, we anticipate that the relationship between self-disgust and negative body image is most pronounced for individuals with relatively strong disgust sensitivity. Theoretically, higher levels of disgust sensitivity are thought to motivate people to avoid stimuli that are expected to evoke disgust [22]. This was shown in an experiment including individuals scoring high and low on contamination fear, where disgust sensitivity was predictive of higher fear and behavioral avoidance of contamination-based tasks [42]. Following this line of reasoning, individuals high in disgust sensitivity are expected to show even higher levels of self-disgust induced avoidance of the own body than individuals scoring low on disgust sensitivity. These avoidance behaviors could contribute to the strengthening of a negative body image by preventing: (1) attention toward and appreciation of attractive aspects of the own body, and (2) habituation to and re-evaluation of self-perceived disgusting body parts (cf. [43]). In other words, access to critical information that could counteract pre-existing negative body image beliefs or help develop positive body image beliefs might become blocked. As a result, avoidance behaviors might contribute to the aggravation of a negative body image over time.

As a first step, we tested this theoretical model in two samples of undergraduate students [44]. Both studies showed that the relationship between disgust propensity and a negative body image indeed could be statistically accounted for by self-disgust. Although disgust sensitivity was found to be positively related to a negative body image, we did not find evidence for the prediction that disgust sensitivity is a moderator of the relationship between self-disgust and a negative body image. Considering the importance of replication studies in psychopathology research, the main aim of the present study was to replicate our prior study and to investigate this theoretical model in a mixed sample of adolescents with and without AN. The inclusion of a mixed (non-)clinical sample was to assure there would be enough variability in our sample. Large variability is recommended when attempting to gain insight into the relationship between continuous variables, because it increases statistiscal power and generalizability of the results [45]. In short, we tested the hypotheses that disgust propensity increases the likelihood for people to have a negative body image by making them more liable to experiencing self-disgust, and that disgust sensitivity moderates the association between self-disgust and negative body image.

In addition, we conducted exploratory analyses to investigate a possible mediating role of body avoidance and body checking in the relationship between self-disgust and negative body image. Both behaviors have been found to be increased in individuals with anorexia nervosa compared to individuals without an eating disorder [46–49]. A study including individuals with eating disorders such as anorexia nervosa indicated that these behaviors either repeatedly alternated or that both behaviors occur simultaneously [50].

Body checking is defined as the frequent and repetitive examination of the own body such as scrutinizing specific body parts or repeatedly studying oneself in the mirror [50]. It has previously been hypothesized that body checking serves as

a way to prevent or limit distress and anxiety as a result of severe preoccupation with body size and shape [51]. Following a similar line of reasoning, body checking may serve as an emotion regulation strategy to prevent or limit feelings of self-disgust. The vigilance towards the own body in response to self-disgust, may then lower the threshold for experiencing rule-violation thereby promoting the feeling of being (too) fat, which in turn could then directly fuel negative attitudes towards the own body and may thus contribute to the persistence of a negative body image (cf. [15]).

Body avoidance involves avoiding reflective surfaces or body exposure by wearing oversized clothes or refusing to be weighed [52]. Self-disgust has been linked to body avoidance in previous quantitative [53] and qualitative research [34]. In the short term, body avoidance may serve as a form of emotion regulation strategy to escape or prevent feelings of self-disgust [54]. However, in the long term, body avoidance could contribute to the strengthening of a negative body image by preventing oneself from the appreciation of attractive aspects of the own body and the habituation and re-evaluation of disgust eliciting body parts.

To sum up, we tested whether (1) self-disgust could statistically account for the relationship between disgust propensity and negative body image; (2) disgust sensitivity moderates the relationship between self-disgust and negative body image; (3) body checking could statistically account for the relationship between self-disgust and negative body image; and (4) body avoidance could statistically account for the relationship between self-disgust and negative body image.

## Method

### Participants

Participants with AN were 64 adolescents ($M_{age}$ = 15.92, $SD_{age}$ = 1.40, range = 14–19) referred for treatment to the Accare department of eating disorders, which is located in the Northern part of the Netherlands. Participants fulfilled DSM-5 criteria for AN restrictive type ($n$ = 47), AN binge purge type ($n$ = 1), AN no subtype known ($n$ = 3), or atypical AN ($n$ = 15). Diagnostic information was obtained from the participants' therapists at Accare, where eating disorder diagnoses are assessed at intake using the child version of the Dutch Eating Disorder Examination (EDE) interview [55]. Atypical AN was diagnosed based on the DSM-5 criteria (i.e., "all of the criteria for anorexia nervosa are met, except that despite significant weight loss, the individual's weight is within or above the normal range"; [1, p. 353]). In addition, we included 62 participants without AN ($M_{age}$ = 16.00, $SD_{age}$ = 1.46, range = 13–19) who were matched on sex, age and educational level with the individuals with AN. No information was collected on race and ethnicity, but there is little ethnic diversity in the Northen parts of the Netherlands, with people typically being from Dutch descent. All participants had female sex, although this was not a selection criterion for this study. Not speaking Dutch was an exclusion criterion for both groups. Additional exclusion criteria for the comparison group were 1) scoring high on eating disorder symptoms (≥ 4; [56,57]) as assessed with the Eating Disorder Examination Questionnaire (EDE-Q; [58]); and 2) scoring outside the healthy weight range (age and gender adjusted BMI ≤ 85 or ≥ 140; [59]). Adolescents of the comparison group were recruited via advertisements that were spread via high schools or via acquaintances of colleagues and students.

### Materials

**Eating disorder examination questionnaire (EDE-Q).** Eating disorder symptoms were assessed with the validated Dutch version of the Eating Disorder Examination Questionnaire (EDE-Q; [57,58]). Adaptations were made to make the language appropriate for adolescents (cf. [60]). The EDE-Q measures the presence of eating disorder symptoms and body concerns in the last 4 weeks (e.g., "Have you had a definite fear that you might gain weight?"), and items are answered on a 7-point Likert scale (0 = *not at all/no days* and 6 = *markedly/every day*). For screening and descriptive purposes, the average score of 22 out of the 28 items (excluding item 13–18) was used to index participants' level of eating disorder symptoms (cf. [55]). Higher scores indicate higher symptom levels (range = 0–6). Internal consistency was excellent (α = .92). Negative body image was measured by combining the weight and shape concern subscales of the

EDE-Q (cf. [44]). The subscales include items assessing the affective-evaluative (e.g., body dissatisfaction, fear of gaining weight) and cognitive-behavioral (e.g., importance of and preoccupation with shape/weight) dimensions of body image [39]. Using the 8 items of the shape subscale and the 5 items of the weight subscale, a mean score of all items combined with a theoretical range of 0–6 was calculated. The combined weight and shape concern subscales showed excellent internal consistency within this study ($\alpha = .90$).

**Disgust propensity and sensitivity scale – revised (DPSS-R).** Disgust propensity and disgust sensitivity were assessed with the Dutch version of the 16-item Disgust Propensity and Sensitivity Scale Revised (DPSS-R; [22]). The DPSS-R consists of 8 items measuring disgust propensity (i.e., the tendency to experience disgust in a wide variety of situations, e.g., "I find something disgusting") and 8 items measuring disgust sensitivity (i.e., how awful do participants consider this disgust experience, e.g., "It scares me when I feel nauseous"). Items are rated on a 5-point scale from 1 (= 'never') to 5 (= 'always') with total scores of each subscale ranging from 8 to 40. The DPSS-R has shown good reliability, convergent validity, and discriminant validity [32]. In the present study, internal consistency was good for both propensity ($\alpha = .88$) and sensitivity ($\alpha = .77$).

**Self-Disgust Eating Disorder Scale (SDES).** Self-disgust was assessed with the Self-Disgust Eating Disorder Scale (SDES; [61]), which was translated to Dutch in collaboration with the original authors. This scale consists of 16 items (e.g., "Parts of my body are gross") which are rated on a 7-point Likert scale ranging from 1 ("strongly agree") to 7 ("strongly disagree"). All items of the questionnaire are listed in S4 Appendix. There are six filler-items, which are not included in the calculation of the total score. To calculate the total score, items 1, 3, 6, 9, 11, 13, and 16 have to be reversed before summing the scores of the ten items. Scores can range from 10 to 70, with higher scores indicating higher levels of self-disgust. The internal consistency of SDES in the present study was excellent ($\alpha = .94$).

**Body checking and body avoidance.** Behaviors regarding avoidance and checking of the own body were assessed with the 27-item Body Checking and Avoidance Questionnaire (BCAQ; [62]). Participants were asked to indicate on a 4-point Likert-scale to what extent each statement applied to them (0 = not at all, 4 = completely applicable). We used the subscales checking behavior (e.g., "I measure the size of my thighs with my hands or with a measuring tape") and avoidance behavior (e.g., "I wear clothes that cover my whole body, even in the summer") consisting of 12 items each. Mean subscale scores are calculated with a scoring range of 0–4, with higher scores indicating higher levels of checking and avoidance behaviors. The internal consistency in the present study was excellent for both subscales (checking: $\alpha = .95$; avoidance: $\alpha = .90$).

**Body mass index (BMI).** Because BMI changes substantially with age, age and sex adjusted BMI was calculated based on self-reported weight and height ([actual BMI/median BMI for age and sex] * 100) [63].

## Procedure

After participants and their parents (when participants were younger than 16) signed informed consent forms, an appointment was scheduled. During this appointment participants filled in the questionnaires at home via an online platform on a computer or laptop. The link to the questionnaire was sent via a secured e-mail. Questionnaires were administered in the following order: DPSS-R, EDE-Q, SDES, BCAQ, demographics including height and weight. Participants also rated food-related vignettes and completed the Body Image State Scale, but both assessments are not relevant to the current study and therefore not reported. Briefly before and after the appointment participants were contacted by the researcher by phone to give instructions and to check whether everything went well. The researcher explained to each participant how they could get help if they kept thinking about the topics in the questionnaire. This included information on a website (www.99gram.nl) and the option of going to their general practitioner or in the case of the patient group, their therapist. All participants received a gift card of €8,00 to thank them for their participation. This study was approved by the medical ethical committee of the University Medical Center in Groningen, the Netherlands (NL.63447.042.17).

## Statistical analyses

To test the hypothesis that disgust propensity increases the likelihood for people to have a negative body image by making them more liable to experiencing self-disgust (H1), a simple mediation analysis was performed with disgust propensity as independent variable, self-disgust as mediator variable, and negative body image as dependent variable (see left panel Fig 2). To test the hypothesis that disgust sensitivity moderates the association between self-disgust and negative body image (H2), a moderated mediation analysis was performed by adding disgust sensitivity as a moderator to the relationship between self-disgust (mediator) and negative body image (dependent variable) (see right panel Fig 2) [64]. Regarding power, results of Fritz and MacKinnon [65] indicated that with the current sample size (N = 126) and 80% power, a medium to medium-small mediation effect could be detected. Here, our previous two studies testing the theoretical model also found medium and medium-small sized mediation effects [44]. To explore the role of body checking and body avoidance in the relationship between self-disgust and negative body image (H3 & H4), two exploratory simple mediation analyses were conducted. Self-disgust was included as independent variable, negative body image as dependent variable in both analysis. One analysis included body avoidance and the other body checking as mediator variable (see left and right panel of Fig 3, respectively). All analyses were conducted in PROCESS version 4.2.

## Results

### Descriptive statistics

An overview of the means and standard deviations, as well as bivariate relations for the variables relevant for the analyses are given in Table 1 and 2, respectively. Correlations per group can be found in S1 Appendix, and EDE-Q global scores per group can be found in S2 Appendix. Notably, compared to the comparison group, mean scores for all variables were

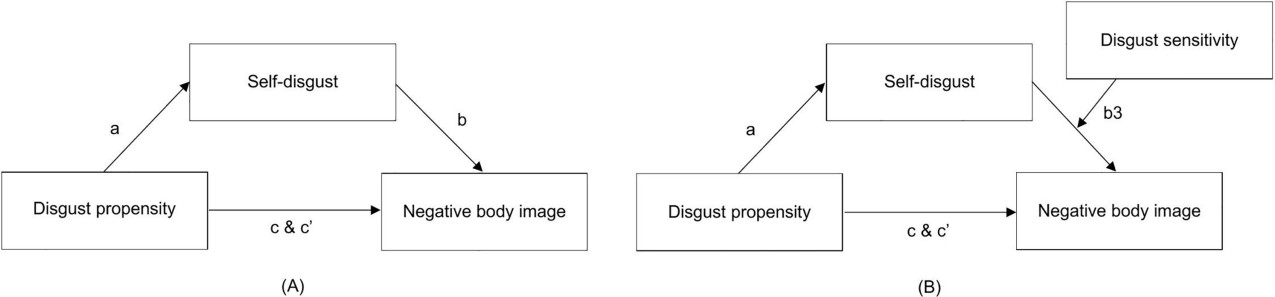

(A)                                                    (B)

**Fig 2. Models representing the hypothesized relationships. (A)** Simple mediation model. **(B)** Moderated mediation model.

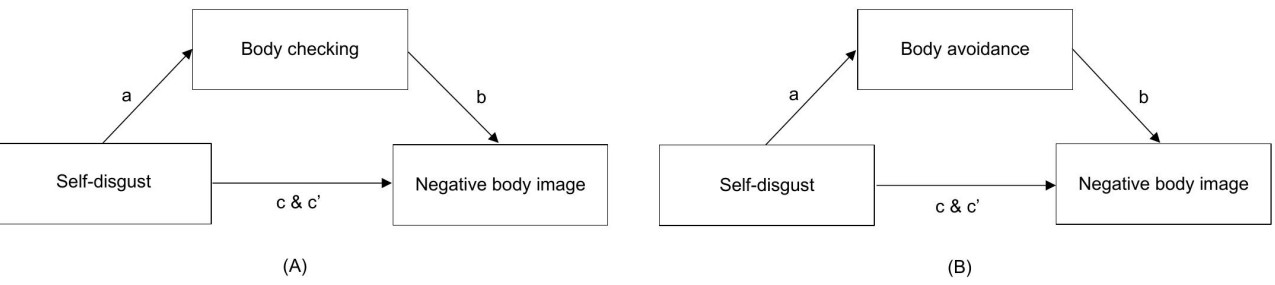

(A)                                                    (B)

**Fig 3. Models representing the relationships investigated in the exploratory analyses. (A)** Body checking as mediator variable. **(B)** Body checking as mediator variable.

**Table 1. Descriptives per group.**

| | Individuals with anorexia nervosa | | Comparison | |
| --- | --- | --- | --- | --- |
| | **Mean (SD)** | **Range** | **Mean (SD)** | **Range** |
| Disgust propensity | 27.73 (5.14) | 18–37 | 19.47 (4.95) | 9–35 |
| Disgust sensitivity | 21.56 (4.48) | 12–31 | 16.19 (5.21) | 8–35 |
| Self-disgust | 45.84 (9.65) | 22–61 | 21.71 (10.07) | 10–53 |
| Negative body image | 4.56 (1.04) | 1.85–6.00 | 1.79 (1.35) | 0.00–5.15 |
| Body avoidance | 2.31 (0.62) | 1.17–3.92 | 1.45 (0.44) | 1.00–3.00 |
| Body checking | 2.78 (0.74) | 1.08–4.00 | 1.52 (0.44) | 1.00–2.92 |

*Note.* Theoretical range of scores: Disgust propensity: 8–40; Disgust sensitivity: 8–40; Self-disgust: 10–70; Negative body image: 0-6; Body avoidance: 0–4; Body checking: 0–4.

**Table 2. Descriptives and bivariate correlations across groups.**

| | Mean (SD) | 2. | 3. | 4. | 5. | 6. |
| --- | --- | --- | --- | --- | --- | --- |
| 1. Disgust propensity | 23.67 (6.52) | .69 | .73 | .66 | .67 | .70 |
| 2. Disgust sensitivity | 18.92 (5.53) | | .61 | .57 | .56 | .54 |
| 3. Self-disgust | 33.97 (15.59) | | | .82 | .67 | .76 |
| 4. Negative body image | 3.20 (1.83) | | | | .65 | .81 |
| 5. Body avoidance | 1.89 (0.69) | | | | | .69 |
| 6. Body checking | 2.16 (0.87) | | | | | |

markedly elevated in the individuals with AN. This is reflected in results of t-tests which showed that levels of disgust propensity ($t$(124) = 9.19, $p$ < .001, $d$ = 1.6) and sensitivity ($t$(124) = 6.21, $p$ < .001, $d$ = 1.1), as well as self-disgust ($t$(124) = 13.74, $p$ < .001, $d$ = 2.4) were significantly higher in individuals with compared to individuals without AN. Furthermore, consistent with the DSM-5 diagnosis for AN, the (adjusted) BMI of individuals with AN ($M$ = 84.46, $SD$ = 10.08) was markedly lower compared to individuals without AN ($M$ = 103.06, $SD$ = 10.28), whereas eating disorder symptoms as measured by the EDE-Q were higher in individuals with AN ($M$ = 3.91, $SD$ = 1.02) compared to individuals without AN ($M$ = 1.33, $SD$ = 1.02).

## Main analysis

Assumption checks for normality, linearity and homoscedasticity did not provide evidence for violations of these assumptions. To facilitate interpretation of the moderated mediation model results and reduce the multicollinearity introduced by its inclusion, variables relevant for this analysis were mean-centered. For consistency, all variables were mean-centered. Outlier inspection showed that for all three models, none of the observations could be considered outliers and thus sensitivity analyses were not considered. To account for potential effects of group differences, we repeated the main analysis with group as covariate. These results were consistent with the effects found in the models reported in the main text. We therefore included the additional analysis in S3 Appendix.

**Simple mediation.** A simple mediation analysis served to determine whether the relationship between disgust propensity and negative body image could be statistically accounted for by self-disgust (H1). The mediation model is depicted in Fig 2 and results are represented in Table 3. Without self-disgust in the model, results showed that disgust propensity had a statistically significant and positive effect on body image (total effect; path c). When entering self-disgust into the model, the strength of this effect reduced to a statistically non-significant effect (direct effect; path c'). Furthermore, results indicated that higher levels of disgust propensity were connected to statistically significantly higher

**Table 3. Results of the mediation for the main analysis.**

| Path/effect | B (SE) | t | p-value | 95%CI |
|---|---|---|---|---|
| c Total effect (DP on NBI) | 2.404(.24) | 9.72 | <.001 | 1.92–2.89 |
| c' direct effect (DP on NBI) | 0.484(.27) | 1.80 | .07 | −0.049–1.02 |
| a (DP on SD) | 1.73(.15) | 11.73 | <.001 | 1.44–2.03 |
| b (SD on NBI) | 1.11(.11) | 9.82 | <.001 | 0.88–1.33 |
| | **Effect** | **Boot SE** | | **Boot CI** |
| ab indirect effect (DP on NBI through SD) | 1.92 | 0.24 | | 1.48–2.42 |

*Note.* DP = disgust propensity, NBI = negative body image, SD = self-disgust.

levels of self-disgust (path a). In turn, higher levels of self-disgust significantly predicted a more negative body image (path b). The indirect effect of disgust propensity on negative body image via self-disgust was large and statistically significant, as indicated by the 95% bootstrap confidence interval that excluded zero (indirect effect; path ab). Together, these results suggest that the effect of disgust propensity on negative body image could be accounted for by self-disgust.

**Moderated mediation.** The simple mediation model was extended to include disgust sensitivity as a moderator of the relationship between self-disgust and negative body image to test our second hypothesis (see Fig 2 and results in Table 4). Results showed that disgust sensitivity did not statistically significantly interact with self-disgust to predict negative body image (path b3). Thus, the current results did not support the hypothesis that the relationship between self-disgust and negative body image is moderated by disgust sensitivity.

## Exploratory analyses

To zoom in on the effect of self-disgust on negative body image, we conducted two subsequent mediation analyses to explore whether the engagement in body checking and/or avoidance of looking at one's body could account for the relationship between self-disgust and negative body image.

**Body checking.** The mediation model is depicted in Fig 3 (left) and results are represented in Table 5. The total effect of self-disgust on body image was statistically significant, with increased levels of self-disgust connecting to an increasingly negative body image (path c). Although this effect was reduced when body checking was added to the model, it remained statistically significant (direct effect; path c'). In addition, results showed that higher levels of self-disgust were associated with statistically significantly higher levels of body checking (path a). In turn, increases in body checking significantly predicted a more negative body image (path b). Lastly, the indirect effect of self-disgust on body image via body checking was statistically large and statistically significant, as indicated by the 95% bootstrap confidence interval

**Table 4. Results of the moderated mediation for the main analysis.**

| Path/effect | B (SE) | t | p-value | 95%CI |
|---|---|---|---|---|
| c' direct effect (DP on NBI) | 0.34(.30) | 1.14 | .26 | −0.25–0.94 |
| b (SD on NBI) | 1.07(.12) | 9.27 | <.001 | 0.84–1.30 |
| b2 (DS on NBI) | 0.32(.31) | 1.04 | .30 | −0.29–0.93 |
| b3 (SDxDS on NBI) | −0.13(.02) | −0.73 | .47 | −0.05–0.022 |
| | **Effect** | **Boot SE** | | **Boot CI** |
| Index of moderated mediation | −0.02 | 0.03 | | −0.08–0.03 |

*Note.* DP = disgust propensity, NBI = negative body image, DS = disgust sensitivity, SD = self-disgust.

**Table 5. Results of the exploratory mediation analysis with body checking being the mediator.**

| Path/effect | B (SE) | t | p-value | 95%CI |
|---|---|---|---|---|
| *c* Total effect (SD on NBI) | 1.25(.08) | 16.02 | <.001 | 1.10–1.41 |
| *c'* direct effect (SD on NBI) | 0.73(.10) | 6.99 | <.001 | 0.53–0.94 |
| *a* (SD on BC) | 0.51(.04) | 13.15 | <.001 | 0.44–0.59 |
| *b* (BC on NBI) | 1.01(.16) | 6.50 | <.001 | 0.70–1.32 |
| | **Effect** | **Boot SE** | | **Boot CI** |
| *ab* indirect effect (SD on NBI through BC) | 0.52 | 0.08 | | 0.38–0.68 |

*Note.* SD = self-disgust, NBI = negative body image, BC = body checking.

that excluded zero (indirect effect; path ab). In sum, these results indicated that the relationship between self-disgust and negative body image could be statistically accounted for by body checking.

**Body avoidance.** The mediation model is depicted in Fig 3 (right) and results can be found in Table 6. The total effect of self-disgust without body avoidance in the model was statistically significant and positive, such that higher levels of self-disgust were associated with a more negative body image (total effect; path c). This effect remained significant when body avoidance was entered in the model (direct effect; path c'). Although results showed that self-disgust statistically significantly predicted higher levels of body avoidance (path a), increases in body avoidance did not statistically significantly relate to a negative body image (path b). The indirect effect of self-disgust on negative body image via body avoidance was not significant, as shown by the 95% bootstrap confidence interval that included zero (indirect effect; path ab). Thus, the results indicate that the relationship between self-disgust and negative body image could not be accounted for by body avoidance.

## Discussion

Low treatment success and high relapse rates of anorexia nervosa [5] point towards a need for a better understanding of the mechanisms underlying its core features. Therefore, the present study aimed to test a theoretical model about how disgust propensity, disgust sensitivity, and self-disgust are connected to a negative body image - one of the key characteristics in individuals with AN. The main findings can be summarised as follows: (1) self-disgust could statistically account for the relationship between disgust propensity and negative body image, and (2) disgust sensitivity did not moderate the relationship between self-disgust and negative body image. In addition, exploratory analyses showed that body checking, but not body avoidance, could statistically account for the relationship between self-disgust and negative body image.

Consistent with our theoretical model and our previous findings when testing the model [44], we found that individuals who have a stronger tendency to experience disgust across a variety of situations and stimuli (i.e., high disgust

**Table 6. Results of the exploratory mediation analysis with body avoidance being the mediator.**

| Path/effect | B (SE) | t | p-value | 95%CI |
|---|---|---|---|---|
| *c* Total effect (SD on NBI) | 1.25(.08) | 16.02 | <.001 | 1.10–1.41 |
| *c'* direct effect (SD on NBI) | 1.18(.12) | 9.77 | <.001 | 0.94–1.42 |
| *a* (SD on BA) | 0.40(.03) | 13.04 | <.001 | 0.34–0.46 |
| *b* (BA on NBI) | 0.19(.23) | 0.81 | .42 | −0.27–0.64 |
| | **Effect** | **Boot SE** | | **Boot CI** |
| *ab* indirect effect (SD on NBI through BA) | 0.07 | 0.09 | | −0.09–0.25 |

*Note.* SD = self-disgust, NBI = negative body image, BA = body avoidance.

propensity), were more likely to report a negative body image. Our prior study showed that in undergraduate students with varying levels of body dissatisfaction, the relationship between disgust propensity and negative body image became less strong when self-disgust was added to the model. In the current mixed sample of adolescents with and without AN, the relationship between disgust propensity and negative body image was even no longer significant when self-disgust was added to the model. This pattern of findings is consistent with the perspective that a higher propensity for disgust increases the likelihood of developing a negative body image via stronger feelings of self-disgust.

In addition to replicating the results of our previous study [44], our findings align with earlier research on the one-to-one associations between self-disgust, disgust propensity, and negative body image. Prior work has shown that individuals with high trait levels of self-disgust, also tend to report elevated disgust propensity [23]. In addition, considering that negative body image is a central feature of AN, the relationship between negative body image and self-disgust may be evident in studies showing that individuals with AN have increased levels of body-directed self-disgust compared to individuals without an eating disorder [25,27,28] as well as in qualitative findings where individuals with AN describe intense feelings of disgust towards the own body [34]. Furthermore, individuals with a negative body image demonstrated increased levels of body-directed self-disgust both when confronted with their own body [31] as well as when recalling autobiographical memories related to their body [35,40,41]. Lastly, a study using trait measures showed that negative body image is positively associated with disgust propensity [44]. The current findings add to the existing literature by investigating self-disgust, disgust propensity and negative body image in one conceptual model.

The present findings did not support the hypothesis that the relationship between self-disgust and negative body image would be especially pronounced in individuals with high disgust sensitivity. This nonfinding is consistent with the pattern of findings in our previous study where we also failed to find evidence for disgust sensitivity being a moderator of the relationship between self-disgust and negative body image [44]. Thus the available evidence cast doubt on the idea that avoidance of the own body induced by experiencing the emotion of self-disgust as unpleasant (i.e., disgust sensitivity) may be critically involved in the persistence of a negative body image. These findings seem in contrast with a study regarding posttraumatic stress disorder (PTSD) among soldiers that did show a moderating effect of disgust sensitivity [43]. In this earlier study, the relationship between peritraumatic disgust and PTSD symptoms at follow-up was especially pronounced in those with relatively high disgust sensitivity. One explanation for not finding a similar pattern in our study could be that self-disgust is an inherently highly negative experience where individual differences in disgust sensitivity might not influence the level of (self-)disgust-induced avoidance. Therefore, the difference with the study of Engelhard et al. (2011) [43] can perhaps be explained by the notion that for external stimuli the affective tone of disgust varies from person to person and thus contributes to the strength of disgust-induced avoidance. Disgust towards the self, however, may be a highly negative experience for everyone, thereby leaving little or no room for individual differences in disgust sensitivity to influence the strength of (self-)disgust-induced avoidance.

The absence of a moderating effect of disgust sensitivity on the relationship between self-disgust and negative body image in both the current mixed (non-)clinical sample and the previous non-clinical sample [44], suggests that our theoretical model requires adjustment in this particular respect. Previous research does indicate a positive, independent relationship between disgust sensitivity and symptoms of anorexia nervosa such as a negative body image [11, 26–28], and therefore it remains relevant to explore if disgust sensitivity should be included somewhere else in the model. Future research is needed to establish whether and how disgust sensitivity contributes to a negative body image.

A closer, exploratory examination of the relationship between self-disgust and negative body image revealed that this relationship could be partly explained by the amount of body checking. This is in line with the idea that body checking may be a response to cope with the aversive feeling of self-disgust, which in turn may contribute to the strengthening of a negative body image through perceived rule violations and the confirmation of self-perceived fatness (cf. [15]). In addition, our findings are in line with previous research showing strong relationships between body checking and AN symptoms [49]. The present study is the first to explore the role of body checking on the relationship between self-disgust and negative

body image. Our findings are consistent with the proposed clinical relevance of addressing body checking behavior in treatment to reduce negative body image in individuals with AN [66]. As the analysis was exploratory, replication is necessary to determine the robustness of these findings.

Although the relationship between self-disgust and negative body image could be statistically accounted for by body checking, exploratory analysis revealed that a similar pattern was absent for body avoidance. Despite that the underlying variables were all strongly related when considered in pairs, our findings do not point to body avoidance as an important factor in the relationship between self-disgust and negative body image. One explanation for this might lie in the nature of the items used to measure body avoidance. The items of the body avoidance subscale of the BCAQ (see S4 Appendix for the items per subscale) primarily include scenarios involving the avoidance of displaying the own body towards others instead of avoidance of the self (e.g., avoiding reflective surfaces or refusing to be weighed; cf. [52]). Therefore, it could still be the case that body avoidance towards the self partially accounts for the relationship between self-disgust and negative body image. We expect that particularly the avoidance of the self, such as looking in the mirror, would prevent oneself from (1) attention toward and appreciation of attractive aspects of one's body and (2) habituation to disgust eliciting body parts. For future research, it seems valuable to develop self-report instruments that can differentiate between body avoidance in the presence versus the absence of others to further disentangle the role of body avoidance.

One of the strengths of the current study is that it is a replication of previous research, thereby adding to the existing literature on (self-)disgust and negative body image. Furthermore, the inclusion of individuals that are in treatment for anorexia nervosa contribute to the generalizability of our results to a clinical population. Alongside its strengths, it is also important to critically evaluate our study. First of all, it should be acknowledged that the measures of self-disgust and negative body image have been suboptimal for capturing the distinct aspects of these constructs. While we did not find evidence of high multicollinearity in our data, the strong correlations between both constructs (see Table 2) may be due to conceptual overlap between the instruments used. For example, the items of the EDE-Q that were used to assess negative body image (e.g., "How often did you feel fat?") could be interpreted as assessing parts of self-disgust. Vice versa, the SDES included items that could be interpreted as assessing parts of negative body image (e.g., "I accept how I look") or even broader self-evaluations ("I accept who I am").

Additionally, it should be acknowledged that we did not differentiate between various aspects of self-disgust. More particularly, self-disgust in AN may not only arise from disgust about the own appearance but may also be driven by certain self-perceived dispositional characteristics and immoral behaviors (cf [29]). We therefore recommend that future research includes more specific measures on different types of self-disgust and negative body image to enable clearer differentiation between constructs. This could also shed light on the most promising treatment target(s).

Another important limitation of the present study is the reliance on cross-sectional data, which prevents us from drawing conclusions about the directions of the relationships. Therefore, it remains uncertain whether disgust propensity might lead to self-disgust and a negative body image, or vice versa. Future research using a longitudinal approach would be helpful in this regard and would allow to test the predictive validity of self-disgust for the development, maintenance and/or relapse of AN. In addition, it may also be helpful to apply an experimental approach in order to gain more insight in the possible causal relationship between self-disgust and negative body image. To test the causal impact of self disgust, it would be necessary to directly target self-disgust to determine whether a decrease in self-disgust indeed leads to a decrease in negative body image. There are various ways in which (body-related) self-disgust could be targeted in treatment.

A first approach to effectively reduce disgust could be via habituation in prolonged exposure therapy [61,62]. Prolonged exposure involves repeated and extended confrontation with the disgust-eliciting stimulus. Considering the inherently disgusting nature of disgust eliciting objects, prolonged exposure is suggested to outperform exposure including expectancy-violation (cf. [15]) (such as used in decreasing fear; e.g., [67]). One way to implement this approach is via body-related mirror exposure, where repeated, therapist-guided confrontation with the own body may promote habituation and reduce

disgust (cf. [66]). Another promising approach is the use of virtual reality, as it enables repeated and prolonged exposure to representations of the own body at a higher weight, with the aim of reducing feelings of body-related disgust toward anticipated weight gain. Beyond prolonged exposure, several other strategies have been proposed, either to directly target self-disgust or to more indirectly enhance body appreciation in treatment. For an overview of these treatment suggestions, see [67]. Nonetheless, it should be acknowledged that the effectiveness of the interventions in reducing body-related self-disgust remains to be established and targeting self-disgust in isolation is inherently challenging within treatment.

## Conclusions

In conclusion, our findings are consistent with the view that self-disgust may be an important factor in the persistence of a negative body image in anorexia nervosa. Individuals who are prone to experiencing disgust in a variety of situations, seem to also be likely to experience higher self-disgust, and in turn are likely to have a negative body image. In addition, the current pattern of findings is consistent with the view that the relationship between self-disgust and negative body image may run, at least in part, through body checking. Our findings contribute to the understanding of the association between negative body image and self-disgust-related mechanisms. Given the limitations of the current cross-sectional and correlational design, future longitudinal and experimental research is needed to help determine if self-disgust is indeed a central factor in the development and persistence of a negative body image and other eating disorder-related problems.

## Supporting information

**S1 File. Regression analyses.**
(PDF)

**S1 Appendix. Bivariate relations per group.** Table A. Bivariate relations in the comparison group. Table B. Bivariate correlations in the treatment group.
(DOCX)

**S2 Appendix. EDE-Q global scores per group.** Table A. EDE-Q global scores per group. Fig A. EDE-Q global score of the treatment group. Fig B. EDE-Q global score of the comparison group.
(DOCX)

**S3 Appendix. Main analyses with group added as covariate.** Table A. Results of the mediation analysis with group added as covariate. Table B. Results of the moderated mediation analysis with group added as covariate.
(DOCX)

**S4 Appendix. Questionnaire items (BACQ, SDES, EDE-Q).**
(DOCX)

## Author contributions

**Conceptualization:** Fleur Boonstra, Peter J. de Jong, Rebecca Schulz.

**Data curation:** Fleur Boonstra, Rebecca Schulz.

**Formal analysis:** Fleur Boonstra, Rebecca Schulz.

**Funding acquisition:** Klaske A Glashouwer.

**Investigation:** Klaske A Glashouwer.

**Methodology:** Peter J. de Jong, Klaske A Glashouwer.

**Project administration:** Klaske A Glashouwer.

**Resources:** Klaske A Glashouwer.

**Supervision:** Klaske A Glashouwer.

**Writing – original draft:** Fleur Boonstra, Klaske A Glashouwer.

**Writing – review & editing:** Fleur Boonstra, Peter J. de Jong, Rebecca Schulz, Klaske A Glashouwer.

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
