## [Decision Letter · Decision Letter 0]

18 Jul 2025

Dear Dr. Boonstra,

Thank you for submitting your manuscript to PLOS ONE. After careful consideration, we feel that it has merit but does not fully meet PLOS ONE’s publication criteria as it currently stands. Therefore, we invite you to submit a revised version of the manuscript that addresses the points raised during the review process.

We look forward to receiving your revised manuscript.

Kind regards,

Irving A. Cruz-Albarran

Academic Editor

PLOS ONE

Journal Requirements:

4. Please include captions for your Supporting Information files at the end of your manuscript, and update any in-text citations to match accordingly. Please see our Supporting Information guidelines for more information: http://journals.plos.org/plosone/s/supporting-information .

Additional Editor Comments:

The reviewers' comments must be addressed in a clear and detailed manner, with each comment being addressed point by point. Follow the journal's guidelines for the review stage.

Reviewers' comments:

Reviewer's Responses to Questions

**Comments to the Author**

1. Is the manuscript technically sound, and do the data support the conclusions?

Reviewer #1: Yes

Reviewer #2: Partly

Reviewer #3: Yes

2. Has the statistical analysis been performed appropriately and rigorously?

Reviewer #1: Yes

Reviewer #2: Yes

Reviewer #3: Yes

3. Have the authors made all data underlying the findings in their manuscript fully available?

Reviewer #1: Yes

Reviewer #2: Yes

Reviewer #3: Yes

4. Is the manuscript presented in an intelligible fashion and written in standard English?

Reviewer #1: Yes

Reviewer #2: Yes

Reviewer #3: Yes

Reviewer #1: • Definition of disgust sensitivity (line 53–54): The current definition 'the extent to which someone finds the emotion of disgust unpleasant' is somewhat incomplete. It does not fully capture the intensity of an individual’s reaction to disgust or the extent to which they find the experience itself distressing or aversive. Clarifying these aspects would provide a more accurate definition. It is not just about finding disgust "unpleasant" but rather about how much one dreads, avoids, or is emotionally affected by feeling disgusted.

• Link Between Eating Disorders and Anorexia Nervosa (AN): The relationship between eating disorders in general and AN specifically is not clearly established in the introduction. While eating disorders are undoubtedly relevant to the discussion, the connection to AN could be made more explicit and better integrated into your theoretical framework. Specifically, what is the relationship between them.

• Bracket Error on Line 65: There is an opening bracket issue: ( instead of [. Please correct this for consistency in your formatting and referencing.

• Role of Disgust Sensitivity as a Moderator: When disgust sensitivity’s role in the model is introduced (line 72 - 77) , it is unclear whether it is intended as a moderator in your model. The figure suggests this role, yet it is not explicitly stated in the text. You make it clearer later

• Statement on Avoidance Behaviors and Negative Body Image (line 85-88): The statement—'These avoidance behaviours could contribute to the strengthening of a negative body image by preventing: (1) attention toward and appreciation of attractive aspects of one’s body, and (2) habituation to and re-evaluation of ‘aversive’ (disgust eliciting) body parts (cf. [42])'—is not well set up. The preceding discussion on disgust sensitivity does not provide sufficient empirical or theoretical justification for the proposed 'blocking effect' of disgust. To strengthen this argument, it would be beneficial to expand on how disgust sensitivity leads to avoidance behaviors and, in turn, reinforces negative body image. This could be achieved by integrating more relevant literature and clearly describing the underlying mechanisms that link disgust sensitivity, avoidance behaviors, and body image disturbance

• Interpretation of Disgust Sensitivity Findings: The statement—'Although disgust sensitivity was found to be positively related to a negative body image, we did not find evidence for the prediction that disgust sensitivity is a moderator of the relationship between self-disgust and a negative body image.'—raises several important questions that need further clarification:

o Does this suggest that disgust sensitivity functions differently in the model than initially expected? If so, how does this align with or contradict prior theoretical assumptions?

o Should the model have been adjusted before conducting this study based on previous findings or pilot data? Given that prior research suggested the model may not perform as expected, what was the rationale for using this approach in the current study? Was it due to limited previous evidence, the need for replication, or differences in the sample population?

• Rationale for Replicating the Study in a Clinical Sample: The statement: "As a next step, the main aim of the present study was to replicate our prior study and investigate this theoretical model in a clinical sample of adolescents with anorexia nervosa.” While the relevance of this study can be intuited, the rationale for replication in a clinical sample should be more explicitly stated.

• Justification for Mediation Analysis (Body Checking) The justification for including body checking as a mediator is not well-developed. While body checking behaviors are associated with AN, the paper does not clearly explain why they mediate the relationship between self-disgust and negative body image. The conceptual link between these variables should be explicitly established before introducing the mediation hypothesis. Additionally, if body checking is integral to the theoretical model, a revised figure integrating it should be included. If its inclusion is exploratory, this distinction should be clarified.

Methodology

• Justification for Self-Disgust in the Context of Eating Disorders: The rationale for measuring self-disgust in individuals with eating disorders could be stronger. Is there a specific type of self-disgust that is most relevant to AN? For example, is eating disorder-specific self-disgust more predictive of negative body image than general self-disgust? Providing this clarification would strengthen the theoretical foundation of your study. This might be something to set up earlier in the document that when referring to self-disgust, you are referring to eating disorder related. Or if this measure is more general than the title, could make that clearer.

Results

• Assumption of Medium-Small Mediation Effect: The study correctly references Fritz & MacKinnon (2007) to justify its statistical power, stating that with the current sample size (N = 126), a medium to medium-small mediation effect could be detected with 80% power. However, I was wondering if the justification could be stronger such as have prior studies in this area found effects of a similar magnitude?

• Justification for Mean Centering: The study applies mean centering to manage multicollinearity, but a stronger justification should be included as research has raised concerns about its appropriateness in certain circumstances (see Echambadi & Hess, 2007; Shieh, 2011; Iacobucci et al., 2016). Given these concerns, the study should clearly justify its use of mean centering and whether other techni

o Echambadi, R., & Hess, J. D. (2007). Mean-Centering Does Not Alleviate Collinearity Problems in Moderated Multiple Regression Models. Marketing Science, 26(3), 438–445. https://doi.org/10.1287/mksc.1060.0263

o Shieh, G. (2011). Clarifying the Role of Mean Centring in Multicollinearity of Interaction Effects. British Journal of Mathematical and Statistical Psychology, 64(3), 462–477. https://doi.org/10.1111/j.2044-8317.2010.02002.x

o Iacobucci, D., Schneider, M. J., Popovich, D. L., & Bakamitsos, G. A. (2016). Mean Centering Helps Alleviate 'Micro' but Not 'Macro' Multicollinearity. Behavior Research Methods, 48, 1308–1317. https://doi.org/10.3758/s13428-015-0624-x

Discussion

• Comparison to Previous Research: The comparison to prior studies (lines 341–345) is somewhat superficial. A more in-depth discussion linking these findings to existing literature would strengthen this section. Specifically, how do these results align with or challenge previous research on the relationship between self-disgust, body image, and eating disorders? Providing a clearer synthesis of prior findings and positioning this study’s contribution within that context would enhance its impact.

• Implications of the Study: The discussion should more explicitly address the theoretical and practical implications of the findings, particularly regarding disgust sensitivity and the role of body checking and body avoidance in the proposed model.

• Relevance of Disgust Sensitivity: Given the non-significant moderation results, should disgust sensitivity still be considered a key variable in this framework? Does its role need to be redefined, or should alternative mechanisms be explored? Clarifying whether disgust sensitivity remains central to the model would provide stronger theoretical coherence.

• Role of Body Checking and Body Avoidance: Are body checking and body avoidance essential components of the model, or do these findings suggest reconsidering their inclusion? If they remain relevant, they should be more clearly integrated into the theoretical framework, either conceptually or through an updated model figure. If their role is more exploratory, this should be explicitly stated.

• The limitations section would benefit from additional discussion on causality and methodological constraints:

• While an experimental approach could provide stronger evidence, ethical concerns in this context should be acknowledged. Is this actually a doable study? Could be clearer on how this would look

• Multicollinearity concerns: The presence of high multicollinearity and conceptual overlap between constructs should be discussed in greater detail. How might this have influenced the findings, and what steps could be taken in future research to address these issues?

Reviewer #2: In the manuscript “Disgust in anorexia nervosa: testing a theoretical model connecting negative body image to disgust propensity, disgust sensitivity, and self-disgust,” the authors sought to build upon prior work by testing a model of disgust in a sample of girls with anorexia nervosa. Strengths include the matched control group and theoretical basis of the study. The manuscript was well-written and a pleasure to read.

I have two primary concerns. The first is that the study was framed, to a certain extent, on the need to test this model in a clinical sample. These aims were only partially achieved because the sample consisted of half a matched control group. While the authors do a good job discussing this, it would be helpful to perhaps alter the title and intro to reflect this. I also think it could be helpful to show correlations by group (or maybe even test whether the bivariate correlations differ between groups) to more directly test whether prior findings in healthy populations extend to those with AN. I realize that the authors are underpowered to run their full analyses in the AN subgroup, only.

My second concern is the discussion of mediation. Mediation isn’t testable in cross-sectional designs (Maxwell & Cole, 2007). However, cross-sectional mediation is statistically the same as testing for a confound/third variable (MacKinnon et al., 2000). I would encourage the authors to avoid mediational language and instead reframe their results as indirect effects or one variable accounting for the relationship of another.

More minor points:

Given the varying approaches in the literature, it would be helpful if the authors could indicate how they defined AAN. The EDE doesn’t have a clear algorithm (to my knowledge).

In the first paragraph, please note that AN occurs in boys and men as well. The authors acknowledge this in the methods, but I think it’s important to note in the introduction as well.

Could the author present EDE-Q Global score Means, SDs, and ranges by group? A cut off of 4 seems quite high for the control group, and it seems likely that many of the AN adolescents were below this cut-off based on clinical experience. Could the authors please clarify why they chose a cut off of 4 for the control group?

Could the authors please clarify which language the questionnaires were administered in?

Coul the authors provide more information about the construct validity of the self-disgust measure used? I had difficulty locating the measure and I’m curious how much it might overlap with self-esteem. The “I accept who I am” sample item does not sound particularly relevant to disgust to me.

Reviewer #3: In this study, the authors investigate the relationship between disgust propensity and negative body image using self-report measures and mediation models. The study is well conducted, and both the methodology and the results are clearly reported and explained. I therefore have only a few comments and suggestions.

1. My main concern relates to the authors’ decision to include both individuals with anorexia nervosa and healthy controls in the same mediation model. Given that patients with anorexia nervosa show significantly higher levels across all key variables in the model (disgust propensity, self-disgust, and negative body image), it is possible that group status may act as a confounding variable. In other words, the observed relationships among the variables could be at least partially driven by group differences rather than by genuine associations between psychological constructs. I encourage the authors to provide a clearer justification for combining the two groups within a single model, and—if feasible—to conduct additional analyses that account for this issue. For example, they could run separate mediation models within each group or include group as a covariate or moderator in the model (e.g., using moderated mediation analysis) to test whether the mediation pathway differs between patients and controls.

2. I found the explanation regarding why body avoidance did not emerge as a significant factor in the relationship between self-disgust and negative body image particularly interesting, especially in relation to the nature of the items included in the scale. Would it be possible to conduct an exploratory model focusing only on the items specifically related to self-avoidance? This could provide more insight into whether this specific component plays a role in the mediation pathway.

3. I would encourage the authors to expand the discussion on the potential therapeutic implications of their findings. In particular, I would be interested to know how they envision an intervention specifically targeting self-disgust, and how such an intervention would differ from those aimed at reducing negative body image.

4. Page 14, line 293, and page 15, line 309: I believe there may be an error in the references to the figures and tables in relation to the models involving body checking and body avoidance. It seems that the labels may have been accidentally reversed.

**Do you want your identity to be public for this peer review?** For information about this choice, including consent withdrawal, please see our Privacy Policy

Reviewer #1: No

Reviewer #2: No

Reviewer #3: **Yes:** Valentina Meregalli

---

## [Author Response · Author response to Decision Letter 1]

9 Oct 2025

Dear Dr. Cruz-Albarran,

On behalf of all authors, I am pleased to submit our revised manuscript PONE-D-24-58634 “Disgust in anorexia nervosa: testing a theoretical model connecting negative body image to disgust propensity, disgust sensitivity, and self-disgust”. Thank you for the positive response and for the opportunity to revise and resubmit.

We appreciate the constructive comments from the reviewers and the editorial team, and have carefully revised the manuscript in light of these suggestions. Detailed responses to each point are provided in the "Response to Reviewers" document.

We look forward to your response.

Yours sincerely,

Fleur Boonstra (Corresponding Author)

---

## [Decision Letter · Decision Letter 1]

11 Nov 2025

Dear Dr. Boonstra,

Thank you for submitting your manuscript to PLOS ONE. After careful consideration, we feel that it has merit but does not fully meet PLOS ONE’s publication criteria as it currently stands. Therefore, we invite you to submit a revised version of the manuscript that addresses the points raised during the review process.

We look forward to receiving your revised manuscript.

Kind regards,

Irving A. Cruz-Albarran

Academic Editor

PLOS ONE

Journal Requirements:

Additional Editor Comments:

Dear Authors,

After reviewing your article, we have determined that your proposal presents valuable results. However, one reviewer indicated that substantial revisions are required. I have included his comments.

The authors have responded to many of the reviewers’ concerns and the inclusion of sensitivity analyses strengthen the manuscript. A few concerns remain.

First, the abstract conclusion and many of the claims in the discussion are not supported by the data- this cross-sectional design cannot support the claim that self-disgust contributes to persistent negative body image in anorexia nervosa.

Mediational language continues to be used throughout the manuscript- I think this is a significant limitation as there are no prospective or causal pathways in the data. Mediation implies causality.

I remain concerned about the high threshold for the EDE-Q scores in the control group- the average level of the AN group (3.91) is consistent with the healthy control range (up to 4). Additionally, I think the EDE-Q may not have been scored correctly. Traditionally, it is scored as the average of the four subscales, not an average of all of the items.

Thank you for including Tables A & B in the sensitivity analyses. The correlations between constructs appear to be meaningfully attenuated within group, even to the point that disgust sensitivity and negative body image have a relatively small relationship in the eating disorder group (r = .25). This suggests that the higher correlations in the combined table (e.g., .57) are driven by diagnostic status. As a result, the relationships in the cross-sectional mediation models are driven by diagnostic status. Though under powered, it would be helpful to see what the results look like within just the AN group. This would provide more confidence that the conclusions reached by the authors are not driven by diagnostic group.

Could the authors please clarify in which mediational pathways they covaried for group? That was not immediately clear from the supplemental tables.

I found lines 376-378 confusing, as I did not think that disgust sensitivity was synonymous with avoidance.

We recommend thoroughly addressing each point to ensure clarity, as failure to do so could result in rejection. Thank you very much for your consideration.

Reviewers' comments:

Reviewer's Responses to Questions

**Comments to the Author**

Reviewer #1: All comments have been addressed

Reviewer #2: (No Response)

Reviewer #3: All comments have been addressed

2. Is the manuscript technically sound, and do the data support the conclusions?

Reviewer #1: Yes

Reviewer #2: No

Reviewer #3: (No Response)

3. Has the statistical analysis been performed appropriately and rigorously?

Reviewer #1: Yes

Reviewer #2: No

Reviewer #3: (No Response)

4. Have the authors made all data underlying the findings in their manuscript fully available?

Reviewer #1: Yes

Reviewer #2: No

Reviewer #3: (No Response)

5. Is the manuscript presented in an intelligible fashion and written in standard English?

Reviewer #1: Yes

Reviewer #2: Yes

Reviewer #3: (No Response)

Reviewer #1: The manuscript is technically sound and analysis appears appropriate. Satisfied with the authors responses to the reviews.

Reviewer #2: The authors have responded to many of the reviewers’ concerns and the inclusion of sensitivity analyses strengthen the manuscript. A few concerns remain.

First, the abstract conclusion and many of the claims in the discussion are not supported by the data- this cross-sectional design cannot support the claim that self-disgust contributes to persistent negative body image in anorexia nervosa.

Mediational language continues to be used throughout the manuscript- I think this is a significant limitation as there are no prospective or causal pathways in the data. Mediation implies causality.

I remain concerned about the high threshold for the EDE-Q scores in the control group- the average level of the AN group (3.91) is consistent with the healthy control range (up to 4). Additionally, I think the EDE-Q may not have been scored correctly. Traditionally, it is scored as the average of the four subscales, not an average of all of the items.

Thank you for including Tables A & B in the sensitivity analyses. The correlations between constructs appear to be meaningfully attenuated within group, even to the point that disgust sensitivity and negative body image have a relatively small relationship in the eating disorder group (r = .25). This suggests that the higher correlations in the combined table (e.g., .57) are driven by diagnostic status. As a result, the relationships in the cross-sectional mediation models are driven by diagnostic status. Though under powered, it would be helpful to see what the results look like within just the AN group. This would provide more confidence that the conclusions reached by the authors are not driven by diagnostic group.

Could the authors please clarify in which mediational pathways they covaried for group? That was not immediately clear from the supplemental tables.

I found lines 376-378 confusing, as I did not think that disgust sensitivity was synonymous with avoidance.

Reviewer #3: (No Response)

**Do you want your identity to be public for this peer review?** For information about this choice, including consent withdrawal, please see our Privacy Policy

Reviewer #1: No

Reviewer #2: No

Reviewer #3: **Yes:** valentina meregalli

---

## [Author Response · Author response to Decision Letter 2]

16 Dec 2025

We have provided detailed responses to each point in the "Response to Reviewers" document.

---

## [Decision Letter · Decision Letter 2]

27 Jan 2026

Disgust in anorexia nervosa: testing a theoretical model connecting negative body image to disgust propensity, disgust sensitivity, and self-disgust

PONE-D-24-58634R2

Dear Dr. Boonstra,

We’re pleased to inform you that your manuscript has been judged scientifically suitable for publication and will be formally accepted for publication once it meets all outstanding technical requirements.

Kind regards,

Irving A. Cruz-Albarran

Academic Editor

PLOS One

Additional Editor Comments (optional):

Based on the reviewers' feedback, we have decided to accept your article. Congratulations on your excellent contribution!

Reviewers' comments:

Reviewer's Responses to Questions

**Comments to the Author**

Reviewer #2: All comments have been addressed

2. Is the manuscript technically sound, and do the data support the conclusions?

Reviewer #2: Yes

3. Has the statistical analysis been performed appropriately and rigorously?

Reviewer #2: Yes

4. Have the authors made all data underlying the findings in their manuscript fully available?

Reviewer #2: Yes

5. Is the manuscript presented in an intelligible fashion and written in standard English?

Reviewer #2: Yes

Reviewer #2: I appreciate the authors' thoughtful responses. They have addressed my concerns. I think thi smanuscript is ready for publication.

**Do you want your identity to be public for this peer review?** For information about this choice, including consent withdrawal, please see our Privacy Policy

Reviewer #2: No

---

## [Editor Report · Acceptance letter]

PONE-D-24-58634R2

PLOS One

Dear Dr. Boonstra,

I'm pleased to inform you that your manuscript has been deemed suitable for publication in PLOS One. Congratulations! Your manuscript is now being handed over to our production team.

Kind regards,

on behalf of

Dr. Irving A. Cruz-Albarran

Academic Editor

PLOS One